# Think in Games: Learning to Reason in Games via Reinforcement Learning with Large Language Models

## Abstract

Large language models (LLMs) excel at complex reasoning tasks such as mathematics and coding, yet they frequently struggle with simple interactive tasks that young children perform effortlessly. This discrepancy highlights a critical gap between declarative knowledge (knowing *about* something) and procedural knowledge (knowing *how to do* something). Although traditional reinforcement learning (RL) agents can acquire procedural knowledge through environmental interaction, they often operate as black boxes and require substantial training data. In contrast, LLMs possess extensive world knowledge and reasoning capabilities, but are unable to effectively convert this static knowledge into dynamic decision-making in interactive settings. To address this challenge, we propose **T**hink-**I**n **G**ames (**TiG**), a novel framework that empowers LLMs to develop procedural understanding through direct interaction with game environments, while retaining their inherent reasoning and explanatory abilities. Specifically, TiG reformulates RL-based decision-making as a language modeling task: LLMs generate language-guided policies, which are refined iteratively through online reinforcement learning based on environmental feedback. Our experimental results show that TiG successfully bridges the gap between declarative and procedural knowledge, achieving competitive performance with dramatically lower data demands compared to conventional RL methods. Moreover, TiG provides step-by-step natural language explanations for its decisions, greatly improving its transparency and interpretability in complex interactive tasks. Furthermore, when adapted to downstream tasks through supervised fine-tuning, the model surpasses larger models containing multiples of its parameters, demonstrating that its acquired reasoning capabilities generalize effectively.

## 1 Introduction

Large language models (LLMs) can write poetry, solve complex math problems, and even generate code (Li et al., 2025; Yu et al., 2024; DeepSeek-AI et al., 2025)—yet they fail at tasks that human children master effortlessly through play. When asked to navigate a simple game environment, these powerful models struggle with basic concepts like spatial reasoning (Dihan et al., 2024; Kolner et al., 2024; Wu et al., 2024) and cause-effect relationships (Ashwani et al., 2024) that emerge naturally from interaction. This paradox reveals a fundamental gap in how AI systems acquire understanding: **the difference between knowing about something and knowing how to do something**.

This gap between declarative (knowing *about* something) and procedural knowledge (knowing *how to do* something) poses a critical challenge for AI development. To bridge it, we need environments where AI systems can safely explore, experiment, and learn from consequences—much like children do through play. Digital games provide precisely such environments, offering controlled yet complex worlds where theoretical knowledge must be transformed into practical understanding through multi-turn interaction with the real world (Ye et al., 2020; Xu et al., 2025).

From classical games like chess and poker (Silver et al., 2016; Southey et al., 2012; Zhuang et al., 2025) to modern video games like Atari (Bellemare et al., 2013), StarCraft II (Vinyals et al., 2017), DOTA II (Font & Mahlmann, 2019), and sandbox games like Minecraft (Wang et al., 2023), these

environments provide rich grounds for measuring and advancing cognitive capabilities of AI including pattern recognition, reasoning, sophisticated planning, and generalization (Xu et al., 2025).

Traditional AI approaches to game solving, such as search algorithms (Silver et al., 2016), handcrafted heuristics (Fernández & Salmerón, 2008; Zhang et al., 2024), and reinforcement learning (RL) (Mnih et al., 2013; Oh et al., 2022), have achieved impressive results in different game environments. However, these methods often rely on extensive domain-specific engineering and require massive amounts of training, limiting their ability to generalize to new or dynamic environments (Xu et al., 2025). Moreover, their decision-making processes are typically opaque (Yang et al., 2025), making it difficult for humans to interpret or trust their decisions. While these systems excel at doing, they are not inherently designed to explain their reasoning, and their capacity is particularly limited in scenarios that require strategic thinking.

The advent of LLMs introduce potential paradigm shift (Wang et al., 2023; 2024). Trained on vast and diverse textual data, LLMs possess broad world knowledge and can generate contextually relevant responses, making them attractive for interactive and reasoning-intensive tasks (Hu et al., 2024b). However, their knowledge is static and derived from text on the web rather than direct interaction with game environments. As our preliminary studies reveal (Appendix C), LLMs often lack the nuanced, procedural understanding required for complex and dynamic games. For example, an LLM can learn strategy such as "avoid pushing the lane too far" from online game walkthroughs. However, LLMs cannot execute this knowledge—the precise definition of "too far" is ambiguous and requires additional understanding that only from actual gameplay experience. Although prompt engineering (Wang et al., 2023) can inject additional game mechanics information, it does not fundamentally transform declarative knowledge into procedural understanding.

This brings us back to our central paradox: traditional RL agents know how but cannot explain why, while LLMs know why but cannot execute how. To bridge this gap, we propose **T**hink-**I**n **G**ames (**TiG**), a novel framework that enables LLMs to develop procedural understanding through direct interaction with the game environment while maintaining their natural ability to reason and explain. Specifically, we reformulate traditional RL decision-making task as a language modeling task: our approach uses an LLM to generate policies in language, which are then refined through online reinforcement learning based on direct interaction with game environments. The game environment provides rewards for each action, and the policy model learns from this feedback while generating step-by-step explanations of its reasoning.

We validate our proposed method in the Honor of Kings (HoK) game. Our experiments demonstrate that TiG bridges the gaps of knowing about something and how to do something. It achieves a deeper understanding of the game mechanics, enabling it to both generate effective strategies and articulate the reasoning process behind these strategies. Moreover, the reasoning capalities can be generalized to downstream applications related to the game, allowing the model to surpass larger models containing multiples of its parameters.

## 2 FORMALIZATION

Our goal is to develop LLMs that are capable of high-level strategic reasoning and decision-making within game environments. While our approach is designed to be broadly applicable across various game genres, this study focuses on Multiplayer Online Battle Arena (MOBA) games as a representative and challenging testbed. MOBA games offer a rich environment for investigating high-level reasoning due to their emphasis on team coordination, long-term planning, and dynamic objectives.

### 2.1 MOTIVATION

To enable LLMs to develop a deep, intrinsic understanding of game mechanics, we draw inspiration from the reasoning processes of expert MOBA players when making decisions. Expert gameplay in MOBA environments is characterized by *macro-level reasoning*, which involves devising and executing team-wide strategies, such as objective control, map pressure[1], and coordinated team maneuvers. Unlike micro-level actions (e.g., precise skill execution), macro-level reasoning prioritizes

---

[1]**Map Pressure**: A strategically advantageous situation that compels opponents into unfavorable positions, facilitating control of the map or capture of major objectives.

long-term objectives and team synergy. Our goal is to equip LLMs with these macro-level reasoning capabilities, fostering a comprehensive understanding of game mechanics and enabling generalization across diverse tasks. We do not focus on micro-level actions, as they require tracking the rapidly changing game states, which is not suited to LLMs given their inherent latency and non-realtime nature.

## 2.2 NOTATION FOR TiG

**Game State Representation.** We formalize the MOBA environment as a sequence of discrete time steps, where each time step $t$ corresponds to a comprehensive *game state* $s_t$. The game state $s_t$ captures all visible information from the primary player's perspective essential for strategic decision-making, including teammate attributes, visible turrets, and map vision data. Hidden information, such as unseen enemy stats, is excluded to maintain realistic gameplay conditions. Formally, a match $m$ is represented as a sequence of $T$ game states $\{s_t\}_{t=1}^T$, where each $s_t$ is a structured representation of the environment at time $t$. To leverage the capabilities of recent LLMs in processing structured data, we represent the game state as a JSON object to facilitate the model understanding the game state. An example JSON object is provided in Appendix Figure 5.

**Macro-level Action Space.** To focus the model on strategic reasoning, we define a finite set of macro-level actions $\mathcal{A} = \{a_1, a_2, \ldots, a_K\}$, where each $a_k$ corresponds to a predefined team objective (e.g., "Push Top Lane", "Secure Dragon", "Defend Base"). This abstraction enables the model to reason about high-level strategies rather than low-level mechanics. In our setting, we define $K = 43$ actions shown in Appendix Table 5, to comprehensively cover the range of meaningful strategies encountered within the gameplay. The finite action space also facilitates subsequent rule-based reward design and evaluation.

**Policy Model.** The policy model in our framework refers to an LLM trained to map game states to macro-level actions. We impose no constraints on the model's architecture, requiring only that it possesses robust instruction-following and structural understanding capabilities, achievable through pre-training on diverse datasets. The policy model is designed to learn effective MOBA strategies and demonstrate a nuanced understanding of game states through our training paradigm.

## 2.3 TASK DEFINITION.

We formalize the task as follows: Given the current game state $s_t$, the model is tasked with predicting the next macro-level action $a_t \subseteq \mathcal{A}$ in natural language that best aligns with optimal team strategy, and provide the corresponding reasoning chains $c_t$ for how to reach the answer. Formally, the model learns a mapping $f : (s_t, i_t) \mapsto (a_t, c_t)$, where $i_t$ denotes any additional context or instructions provided to the LLM. This $(s_t, i_t) \mapsto (a_t, c_t)$ prediction task encourages the LLM to analyze the current environment, extract salient information, and predict the most appropriate macro-level action using natural language. As the game state representation can be lengthy and information-rich, the model have to actively explore the environment, identify relevant features, and make informed decisions. The prompt template is shown in Table 1, where the placeholders will be replaced with real data during training and inference time.

## 3 METHOD

To tackle the challenges of traditional RL agents know how but cannot explain why, while LLMs know why but cannot execute how. Our new framework **T**hink-**I**n **G**ames (**TiG**), enables LLMs to develop procedural understanding through direct interaction with the game environment, while maintaining their natural ability to reason and explain. By grounding the learning process in environmental rewards and state transitions, our approach fosters a deep, intrinsic understanding of game mechanics, such as positioning and risk assessment, moving beyond brittle pattern-matching of existing strategies.

In this section, we first outline our systematic data collection and sampling strategies from real game-plays. We then detail our reinforcement learning framework, including the GRPO algorithm and the design of our rule-based reward function.

Given the real-time in-game situation information of a Honor of Kings match, please analyze the current status and provide decision-making objectives.

**In-game situation information:** <game_state> </game_state>

**Key points for consideration:**

1. Situational Awareness: tracking unit states (health, items, cooldowns), map states (minion waves, structures), and vision control (ward coverage, fog of war).

2. Strategic Composition: leveraging team synergies, identifying key power spikes (levels, items), and defining clear roles and responsibilities.

3. Real-Time Analysis: assessing game momentum, predicting opponent intent, and performing the tactical calculus of engagements and resource trades.

4. Scenario Handling: adapting to high-stakes objectives (Tyrant, Overlord), and executing or countering specialized mechanics and strategies.

**Suggested action:** Select the most suitable action from the candidate options. Place the decision/thinking process in <think> </think>, and place the sugggested action in <answer> </answer>. The candidate options are <action_candidates> </action_candidates>

**Table 1:** Prompt Template for TiG. <game_state> </game_state> will be replaced with the real game state during training and inference and <action_candidates> </action_candidates> will be replaced with the predefined action sets of $\mathcal{A}$ defined in Appendix Table 5.

### 3.1 DATASET COLLECTION

Our dataset were sampled from anonymized records of real game matches, where neither user identifiers nor any personally identifiable information were collected to safeguard player privacy. To ensure balanced representation, we maintain an equal ratio of wins and losses, and only include matches played by users above a skill threshold.

#### 3.1.1 DATA SAMPLING STRATEGY

For each match, we first extract the full sequence of game states, and label the main player's action as the ground truth for each game state. Since state transitions could lead to inconsistent or sparse action labels, we develop a relabeling algorithm to densify and smooth the annotation sequence. The relabeling algorithm (Section 3.1.2) ensures that each game state with a macro-level action label. After relabeling, we employ a random sampling strategy to select one frame per minute of gameplay to ensure the diversity of our training data.

#### 3.1.2 RELABELING ALGORITHM

**Priority-based Hierarchy of Macro-level Action.** During the game-play, actions exhibit varing degrees of priority. For example, critical objectives such as "Tyrant or Overlord" and "Team Fight" should be prioritized. We formalize action priority as a function of criticality, temporal window, and overall game impact: $\text{Priority}(a_t) = f(\text{criticality}, \text{time\_window}, \text{game\_impact})$. We provide the hierarchy based on expert human player knowledge as shown in Table 5.

**Relabeling Algorithm.** Since game state transitions could lead to inconsistent or sparse action labels, we develop a relabeling algorithm to densify and smooth the annotation sequence. Formally, given a sequence of game states from gameplay, our labeling algorithm first propagate the detected action label backward to preceding unlabeled frames within a window of $L_{\text{fill}}$ frames. This ensures that each game state is associated with a relevant macro-level action, even if the original annotation was sparse. After backward filling, some frames may be associated with multiple overlapping actions due to the temporal proximity of different actions. To resolve conflicts and ensure that the most important action is represented, we leverage the predefined priority hierarchy (also shown in Table 5). Within a window of $L_{\text{overwrite}}$ frames, if multiple actions overlap, we overwrite lower-priority action labels with higher-priority ones according to the hierarchy. This process guarantees that, at any given frame, **the label reflects the most critical macro-level action occurring at that time**.

By explicitly incorporating action priority into the relabeling process, our algorithm produces a dense and consistent sequence of game states, where each game state is labeled with the most contextually important macro-level action. This results in a robust training signal for downstream learning tasks.

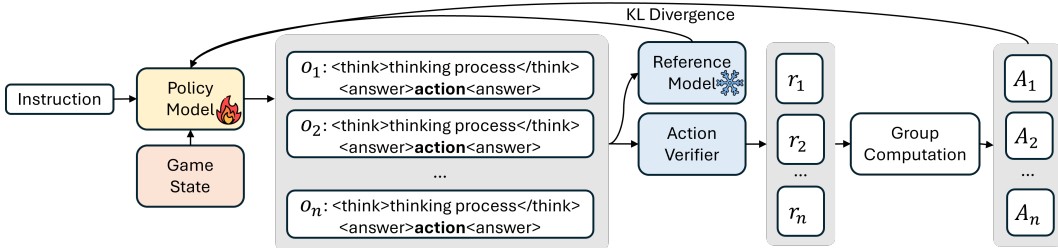

**Figure 1: Demonstration** of GRPO training with Game State. 🔥 refers to trained models, and ❄ refers to frozen models. Given the current game state, the model is asked to predict the proper action, and provide the thinking process as the analysis of why consider this action. We then compare the predicted action with ground-truth values using a rule-based verifier to update the policy model. This process enables the model to perform decision making within the game environment and refining its decision-making accordingly.

## 3.2 REINFORCEMENT LEARNING WITH GAME STATE

To enable effective learning of strategic reasoning in game environments, we adopt a reinforcement learning (RL) framework that directly optimizes the policy model using feedback from game state-action pairs. Specifically, we employ *Group Relative Policy Optimization* (GRPO) (Shao et al., 2024), an online RL algorithm designed to maximize the advantage of generated completions while constraining policy divergence from a reference model. We provide further analysis on why we use GRPO for our task and explain how our adaptation differs from the original version in Appendix D.

**GRPO Formalization.** We formalize our training process of TiG using GRPO below. Let $q$ denote a sampled prompt (*e.g.*, a game state $s_t$ and context $i_t$), and let $\{o_1, o_2, \ldots, o_G\}$ be a group of $G$ completions generated by the current policy $\pi_\theta$. For each completion $o_i$, a reward $r_i$ is computed using a rule-based reward function (see Reward Modeling below). The group-relative advantage for each completion is then calculated as:

$$\hat{A}_{i,t} = \frac{r_i - \text{mean}(r)}{\text{std}(r)}, \tag{1}$$

where $\text{mean}(r)$ and $\text{std}(r)$ denote the mean and standard deviation of rewards within the group. This normalization ensures that the advantage reflects the relative quality of each completion.

To regularize policy updates, we estimate the token-level Kullback-Leibler (KL) divergence between the current policy $\pi_\theta$ and a reference policy $\pi_{\text{ref}}$ using Schulman's approximator (Schulman et al., 2017):

$$\mathbb{D}_{\text{KL}}\left[\pi_\theta \| \pi_{\text{ref}}\right] = \frac{\pi_{\text{ref}}(o_{i,t}|q, o_{i,<t})}{\pi_\theta(o_{i,t}|q, o_{i,<t})} - \log \frac{\pi_{\text{ref}}(o_{i,t}|q, o_{i,<t})}{\pi_\theta(o_{i,t}|q, o_{i,<t})} - 1. \tag{2}$$

The overall GRPO objective is to maximize the expected group-relative advantage while penalizing excessive policy drift. The loss function is defined as:

$$\mathbb{L}_{\text{GRPO}}(\theta) = -\frac{1}{\sum_{i=1}^{G}|o_i|} \sum_{i=1}^{G}\sum_{t=1}^{|o_i|}\left[\min\left(\frac{\pi_\theta(o_{i,t}|q, o_{i,<t})}{\pi_{\theta_{\text{old}}}(o_{i,t}|q, o_{i,<t})}\hat{A}_{i,t}, \text{clip}\left(\frac{\pi_\theta}{\pi_{\theta_{\text{old}}}}, 1-\epsilon, 1+\epsilon\right)\hat{A}_{i,t}\right) - \beta\mathbb{D}_{\text{KL}}\right], \tag{3}$$

where the clipping operator $\text{clip}(\cdot, 1-\epsilon, 1+\epsilon)$ constrains the update magnitude, and $\beta$ controls the strength of KL regularization. This formulation enables token-level optimization with sequence-level rewards, facilitating efficient and stable learning.

**Reward Modeling.** The reward function serves as the primary training signal, guiding the optimization process in RL. Following the success in Deepseek-R1 (DeepSeek-AI et al., 2025) using rule-based reward, we adopt a similar rule-based reward system that consists solely of **final outcome rewards**, which assess the correctness of the model's responses. Formally, given a predicted action $\hat{A}_t$ at time step $t$ and the corresponding ground truth action $A_t^*$ obtained from the replay data, the reward $r_t$ is defined as:

$$r_t = \begin{cases} 1, & \text{if } \hat{A}_t = A_t^*, \\ 0, & \text{otherwise.} \end{cases} \tag{4}$$

The reward is assigned 1 if the predicted action matches the ground truth action; otherwise, the reward is 0. This binary reward encourages the model to generate action predictions that closely match real player behavior while penalizing overly verbose or irrelevant outputs. We do not incorporate format rewards, as our learned model already demonstrates strong structural adherence. Furthermore, following Deepseek-R1 (DeepSeek-AI et al., 2025), we avoid training neural reward models. The decision is motivated by the sensitivity of LLMs to specific forms of rewards in large-scale RL training process, as well as the additional computational cost and complexity introduced by retraining these models.

## 4 EXPERIMENTS

In this section, we first illustrate the experiment setup for our experiments, including datasets, environments and the baseline models for the comparison (Section 4.1). Then we discuss the training details of our method (Section 4.2), and provide the detailed analysis of the main results (Section 4.3). We also present ablation studies, error analysis (Section 4.4) and case studies (Section 4.5) to further analyze the effectiveness of the proposed method.

### 4.1 EXPERIMENT SETUP

**Environment.** All experimental results are obtained on four servers with 8 NVIDIA H20 (96 GB) GPUs. For SFT, we use the Megatron-LM (Shoeybi et al., 2019) training platform. For online RL, we use the OpenRLHF (Hu et al., 2024a) training platform.

**Datasets.** We evaluate our models in two settings. First, within game environments, we utilize complex scenarios sampled from the HoK game, as described in Section 3.1. To prevent data leakage, we re-sample a subset of examples and slightly modify the output format. This allows us to assess whether the trained models can generalize to new tasks beyond their training distribution. In this setting, the model is provided with the current game state in JSON format and a finite action space; it is tasked to select the appropriate action and generate a corresponding reasoning process. Given the need for detailed analysis of the game state and a strategic understanding of game mechanics, we employ expert human evaluators (experienced game players) to assess the quality of the model's outputs. To further verify whether our models sacrifice their native language understanding and reasoning capabilities, we further evaluate their performance on diverse standard benchmarks: Ape210K (Zhao et al., 2020), MMLU (Hendrycks et al., 2021), CEval (Huang et al., 2023), School-Chinese (lanhin, 2018), BBH (Suzgun et al., 2023), IfEval (Zhou et al., 2023) and CharacterEval (Tu et al., 2024). The detailed description of the benchmarks and the public data links can be found in Appendix B.1.

**Baselines.** We consider LLMs of various scales as baselines. We include several other publicly available models in our experiments: Qwen-2.5-7B-Instruct, Qwen-2.5-14B-Instruct, Qwen-2.5-32B-Instruct, Qwen-3-14B-Instruct, and Deepseek-R1. All model checkpoints are accessible via Hugging Face[2]. For training, we set $\text{prompt\_max\_len} = 8192$ and $\text{generate\_max\_len} = 2048$.

### 4.2 TRAINING DETAILS

We follow the insights from Deepseek-R1 (DeepSeek-AI et al., 2025) to employ multi-stage training that combines supervised fine-tuning (SFT) and reinforcement learning (RL) to enhance the capabilities of our language models. Specifically, SFT helps improve foundational language understanding and reasoning of our models, while online RL teaches the models to efficiently explore and select the most effective solutions through trial and error.

For the SFT stage, we distill the training data from Deepseek-R1, which demonstrates strong reasoning capabilities in game environments and can thoroughly analyze the game states based on its pre-existing knowledge. We consider this makes the training data a valuable resource for training smaller models to acquire deep reasoning capabilities. For the online RL stage, we use real gameplay data collected as described in Section 3 and train the models with the GRPO algorithm (Shao et al., 2024). Due to computational constraints, we vary the number of training steps across models: Qwen2.5-14B is

---

[2] https://huggingface.co/models

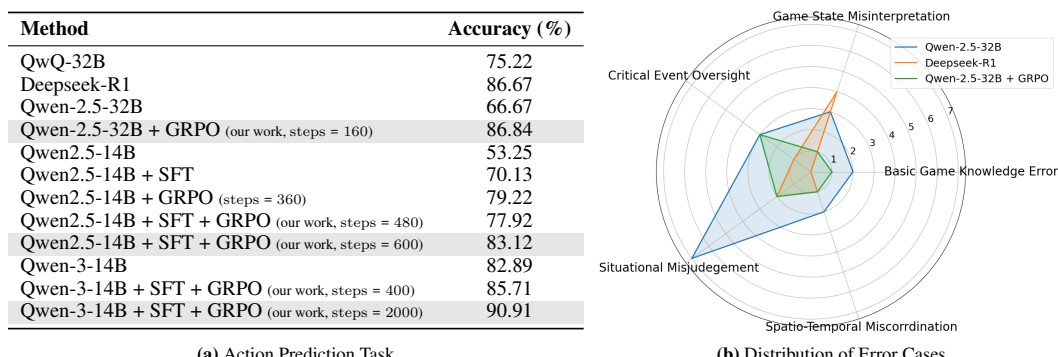

| Method | Accuracy (%) |
|---|---|
| QwQ-32B | 75.22 |
| Deepseek-R1 | 86.67 |
| Qwen-2.5-32B | 66.67 |
| Qwen-2.5-32B + GRPO (our work, steps = 160) | 86.84 |
| Qwen2.5-14B | 53.25 |
| Qwen2.5-14B + SFT | 70.13 |
| Qwen2.5-14B + GRPO (steps = 360) | 79.22 |
| Qwen2.5-14B + SFT + GRPO (our work, steps = 480) | 77.92 |
| Qwen2.5-14B + SFT + GRPO (our work, steps = 600) | 83.12 |
| Qwen-3-14B | 82.89 |
| Qwen-3-14B + SFT + GRPO (our work, steps = 400) | 85.71 |
| Qwen-3-14B + SFT + GRPO (our work, steps = 2000) | 90.91 |

(a) Action Prediction Task.  (b) Distribution of Error Cases

**Figure 2:** (left) Action Prediction Task, (right) Distribution of the Error Cases across different models. The definition of error cases can be found in Table 4.

trained for up to 700 steps, Qwen-2.5-32B for up to 160 steps, and Qwen3-14B—showing consistently strong performance—is trained for over 2,000 steps to better observe training dynamics.

### 4.3 MAIN RESULTS

For our experiments, we explore different combinations of multi-stage training: (1) **GRPO**: Train the base model using GRPO only without applying SFT training. (2) **SFT**: Train the base model using SFT training dataset. (3) **SFT + GRPO**: Start by training the base model with SFT, then apply the GRPO algorithm to further train the model to improve its reasoning abilities.

The main results can be found in Table 2a, from which we draw several key findings. **First**, multi-stage training—particularly the combination of SFT and GRPO—leads to substantial improvements in model performance across different model sizes. For example, Qwen-2.5-32B improves from 66.67% (base) to 86.84% with GRPO, while Qwen2.5-14B increases from 53.25% (base) to 83.12% after sequential application of SFT and GRPO. This demonstrates the effectiveness of our approach in enhancing complex reasoning abilities. **Second**, our training strategy enables smaller models to rival or even surpass much larger models. Notably, Qwen-3-14B with SFT and extended GRPO training (2000 steps) achieves 90.91% accuracy, outperforming Deepseek-R1 (86.67%), which is an order of magnitude larger in parameter count. This highlights the efficiency and scalability of our method. **Third**, reinforcement learning via GRPO is a key driver of reasoning improvement. The introduction of GRPO, either alone or following SFT, consistently yields significant accuracy gains. For instance, Qwen-2.5-32B + GRPO achieves a 20-point increase over the base model, and Qwen2.5-14B + GRPO (79.22%) outperforms SFT only. These results confirm that GRPO is particularly effective for boosting the reasoning capabilities of language models.

### 4.4 ANALYSIS

**Generalization of TiG.** To verify TiG's generalizability on other tasks, we propose another dataset collected from real gameplay in question-answering format (denoted as TiG-QA). In the TiG-QA task, the model is given the game state and an open-ended user question, and is asked to generate a comprehensive answer grounded in the game context. As shown in Table 2 (TiG-QA Task). While Deepseek-R1 still shows some advantages on certain general capability questions related to game states, as seen in Table 2, we believe this is because these questions are less tied to the game environment and more like open-ended queries that rely on prior knowledge from the web rather than interaction with the game. This is an area where Deepseek-R1 excels.

**Error Analysis.** We conduct further error analysis in Figure 2b, where we categorize errors occured in TiG-QA based on the definitions in Appendix Table 4. We find that our method generally outperforms the base model and achieves results comparable to Deepseek-R1 (671B). Considering that our model only has 32B parameters, this highlights the effectiveness of our approach.

| Model | Game-state Strong Related (e.g., decision) | | | Game-state Weak Related (e.g., items) | | |
|---|---|---|---|---|---|---|
| | 0 | 1 | 2 | 0 | 1 | 2 |
| Deepseek-R1 | 17.14% | 25.71% | 57.14% | 3.70% | 16.67% | 79.63% |
| Qwen-2.5-32B | 31.43% | 38.57% | 30.07% | 34.78% | 26.09% | 39.13% |
| Qwen-2.5-32B + GRPO (our work, steps = 160) | 21.43% | 38.57% | 40.00% | 28.89% | 33.33% | 37.78% |

**Table 2:** Model performance on board-related tasks. Numbers indicate count (percentage) of correct responses. Scoring: 0 = Incorrect, 1 = Partially Correct, 2 = Correct.

## 4.5 Case Studies

To provide a comprehensive evaluation of TiG's capabilities, we conduct a detailed case study based on the real-time game scenario depicted from Figure 3 to Figure 14. They qualitatively showcase that TiG advanced capacity for deep, context-aware reasoning and its ability to translate complex game states into actionable, natural language guidance for the player.

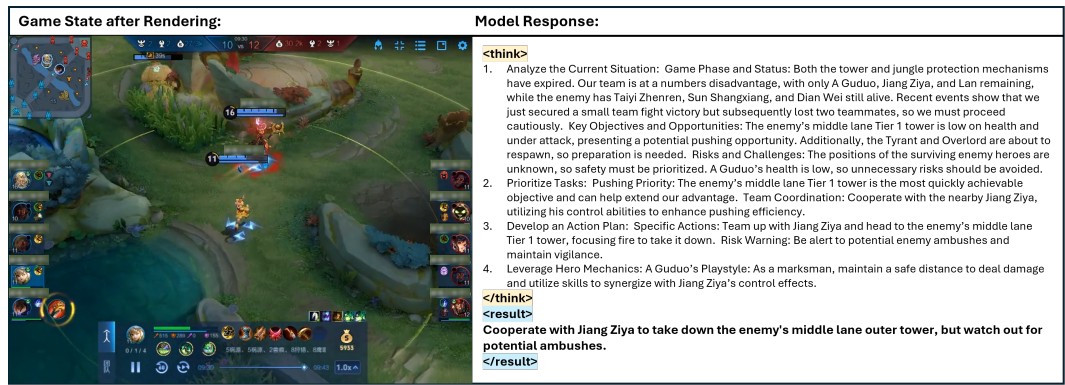

**Figure 3:** One of the cases of TiG. <think> </think> refers to the thinking process of model output, and <result> </result> refers to the model guidance to the main player in natural language.

As illustrated in the Figure 3, the scenario involves the main player controlling the hero A Guduo, who is pushing the mid-lane with a teammate, Jiang Ziya, against a weakened enemy tier-one tower. The model's internal reasoning process, displayed on the right, is methodical and multi-faceted: (1) **Situation Analysis**: The model first performs a holistic assessment of the game state. It identifies that the match has progressed beyond the early game, noting that the "defensive tower and jungle protection mechanisms have expired." However, it mistakenly accounts for the team's numerical disadvantage, as for two teams, they all have three heroes remained. It then analysis that there is a recent skirmish and highlights the low health of the enemy mid-lane tower as a primary opportunity. Crucially, it also identifies key risks, such as the unknown positions of the enemy heroes and A Guduo's low health, which necessitates caution. (2) **Objective Prioritization**: Based on its analysis, the model prioritizes objectives. It determines that destroying the mid-lane tower is the most immediate and achievable goal to capitalize on the current momentum and expand the team's advantage. It emphasizes the importance of teamwork, specifically coordinating with the nearby Jiang Ziya to leverage his crowd-control abilities for a safer and more efficient push. (3) **Strategy Formulation**: The model then synthesizes this analysis into a concrete action plan. The core directive is to "join Jiang Ziya at the enemy mid-lane tier-one tower and focus fire to bring it down." This strategy is coupled with a critical risk mitigation warning: "Be aware that enemy heroes may be lying in ambush; maintain vigilance." (4) **Hero-Specific Playstyle Integration**: The reasoning demonstrates an understanding of hero roles, advising that A Guduo, as a marksman, should "maintain a safe distance for output" and utilize her skills in tandem with Jiang Ziya's control effects.

Finally, the model distills this intricate chain of reasoning into a single, clear, and concise instructional output for the player: "Jointly push down the enemy mid-lane tier-one tower with Jiang Ziya; be mindful of a potential enemy ambush." This case study effectively showcases that TiG is not merely reactive but engages in a proactive, strategic decision-making process within the game environment.

It balances opportunity with risk, incorporates hero-specific knowledge, and ultimately provides guidance that is both **tactically sound** and **immediately executable for a human player**.

## 5 RELATED WORK

**Game Understanding of LLMs.** While large language models (LLMs) excel at language-based reasoning, effectively applying them to games remains challenging. This difficulty stems from their reliance on static pre-training data and a lack of true environmental grounding (Hu et al., 2024b). The main challenges are as follows: (1) *Contextual grounding*: LLMs struggle to interpret dynamic and evolving game states, making it difficult for them to make consistent and accurate decisions based on real-time information from the game environment (Hu et al., 2024c); (2) *Symbolic precision*: LLMs can misinterpret subtle differences in game terminology or item attributes—such as confusing a "dagger" with a "shortsword"—which can disrupt their interaction with the game engine (Southey et al., 2012); and (3) *Long-term planning and memory*: Many games require strategic reasoning over extended time horizons, a task that remains difficult for LLMs due to their limited memory and planning capabilities (Silver et al., 2016; Vinyals et al., 2017; He et al., 2025). In this paper, we address these challenges by bridging the gap between LLMs and game environments, enabling LLMs to develop experiential understanding through direct interaction while preserving their natural strengths in reasoning and explanation.

**Role of RL in LLMs.** Recent advances in LLMs have highlighted the crucial role of RL in aligning model outputs with human preferences (Sui et al., 2025; Jin et al., 2025). While pre-training LLMs on vast text corpora enables them to generate fluent and grammatically correct text, this alone is insufficient for ensuring that models are helpful, harmless, and aligned with user expectations. Supervised fine-tuning (SFT) can improve structure but often fails to guarantee factual accuracy or mitigate biases. RL, particularly through RL from human feedback (RLHF) (Ouyang et al., 2022), addresses these limitations by training a reward model based on human preferences to guide further policy optimization, commonly using Proximal Policy Optimization (PPO) (Schulman et al., 2017). However, PPO introduces complexity due to its reliance on multiple optimization rounds and the need for a separate reward model. To simplify this process, methods such as Direct Preference Optimization (DPO) (Rafailov et al., 2023) and SimPO (Meng et al., 2024) have been proposed. These approaches reframe the problem as a classification task between preferred and rejected responses, eliminating the need for a separate reward model by leveraging preference data directly.

More recently, Group Relative Policy Optimization (GRPO) (Shao et al., 2024) has emerged as a flexible alternative for obtaining reward signals. Unlike PPO, GRPO does not strictly require a reward model; instead, it can incorporate reward signals from any function or model capable of evaluating response quality. For example, one could use a length function to reward concise answers, a mathematical solver to verify solution correctness, or a factuality checker to encourage more accurate responses. This flexibility makes GRPO particularly versatile for a wide range of alignment tasks. Despite these advancements, the application of RL-based alignment methods—especially in game-related domains—remains an open area for further exploration.

## 6 CONCLUSION

In this work, we introduced **Think-In-Games (TiG)**, a novel framework that empowers LLMs to acquire procedural knowledge through direct interaction with game environments, while retaining their natural strengths in reasoning and explanation. By reformulating RL as a language modeling task, TiG enables LLMs to generate interpretable, language-guided policies that are refined via online feedback. Our experiments demonstrate that TiG not only bridges the gap between knowing about and knowing how to do, but also achieves competitive performance with significantly reduced data and computational requirements compared to traditional RL approaches. Furthermore, TiG produces step-by-step explanations for its decisions, enhancing transparency and interpretability in complex, interactive tasks. We believe this framework opens new avenues for developing AI agents that can both act effectively and explain their reasoning in dynamic environments.

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

## A  RESPONSE LENGTH VS. REWARDS.

We illustrate how rewards and response lengths change during RL training process in Figure 4. For Qwen2.5-14B and Qwen-2.5-32B, response length follows a pattern of decreasing, then increasing, and finally stabilizing, which aligns with their overall performance trends. In contrast, Qwen-3-14B's response length steadily increases throughout training. This may be because Qwen-3-14B is designed to support deeper thinking and benefits from scaling laws—generating more tokens tends to improve its capabilities.

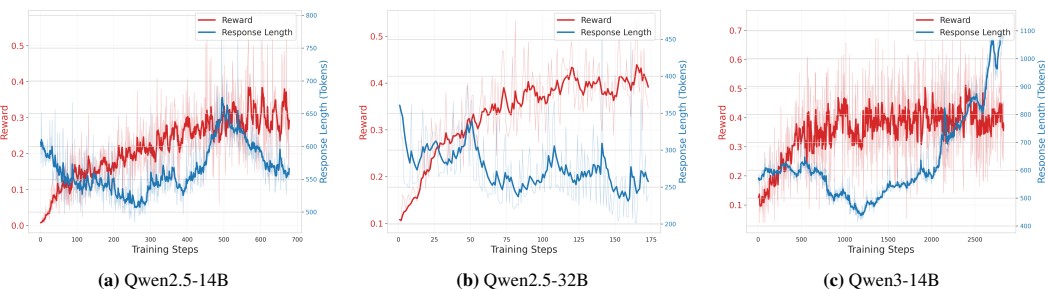

(a) Qwen2.5-14B     (b) Qwen2.5-32B     (c) Qwen3-14B

**Figure 4:** Demonstration of Rewards & Response Length change during the RL training process.

## B  EXPERIMENT SETUP

### B.1  DETAILED DESCRIPTIONS OF THE BENCHMARKS

We evaluate our models on several different benchmarks that target on various capabilities of large language models, including reasoning (math), memorization, domain-specific knowledge (subject examination), dialogue, logical reasoning and the instruction following, etc. The details of the benchmarks are as follows:

- **Ape210K** (Zhao et al., 2020): A large-scale and template-rich math word problem dataset. For our experiments, we randomly sample 200 examples from the test set.

- **MMLU** (Hendrycks et al., 2021): A comprehensive benchmark covers knowledge from 57 subjects across STEM, the humanities, the social sciences, etc. It ranges in difficulty from an elementary level to an advanced professional level, and it tests both world knowledge and problem solving ability. We sample the first 50 examples from each subject and collect $50 * 57 = 2850$ cases for our experiments.

- **CEval** (Huang et al., 2023): Similar to MMLU, CEval is a Chinese-language benchmark comprising 52 subtasks across four categories: STEM, social sciences, humanities, and others. We consider it as an additional testbed to evaluate language mixing challenges (DeepSeek-AI et al., 2025).

- **School-Chinese** (lanhin, 2018): This benchmark assesses memorization capabilities of LLMs on classical Chinese poems by requiring the model to predict subsequent content given introductory text. We collect these datasets manually from the public data repository and construct a benchmarks covering 269 samples.

- **BBH** (Suzgun et al., 2023): A subset of the BIG-Bench (Srivastava et al., 2023) focusing on a suite of 23 challenging tasks that require multi-step reasoning. It is widely regarded as a standard evaluation set for assessing the logical reasoning abilities of language models.

- **IfEval** (Zhou et al., 2023): A standard benchmark for evaluating instruction following capabilities of LLMs. It contains approximately 500 verifiable instructions, such as "write more than 400 words" or "mention the keyword 'AI' at least three times," which can be automatically checked using heuristics.

- **CharacterEval** (Tu et al., 2024): A Chinese benchmark for evaluating role-playing conversational abilities. It includes 1,785 multi-turn dialogues and 23,020 examples featuring 77 characters drawn from Chinese novels and scripts.

## B.2 Evaluation on General Benchmarks

| Model | Math | Memorization | Subject Exam | | Dialogue | Logical Reasoning | Instruction Following |
|---|---|---|---|---|---|---|---|
| | ape_210k | SchoolChinese | MMLU | Ceval | CharacterEval | BBH | IfEval |
| Qwen2.5-14B | 79.0 | 95.7 | 76.2 | 76.7 | 3.05 | 60.7 | 64.8 |
| Qwen2.5-14B+SFT+GRPO | 78.8 | 95.8 | 76.3 | 76.2 | 3.04 | 61.3 | 64.9 |
| Qwen3-14B | 93.5 | 91.6 | 80.3 | 83.1 | 3.01 | 65.8 | 65.8 |
| Qwen3-14B+SFT+GRPO | 94.1 | 91.3 | 80.4 | 82.8 | 3.01 | 66.9 | 65.3 |
| Qwen2.5-32B | 84.0 | 98.1 | 83.2 | 87.7 | 3.13 | 67.5 | 85.8 |
| Qwen-2.5-32B + GRPO | 85.0 | 97.9 | 83.5 | 87.4 | 3.13 | 69.7 | 85.3 |

**Table 3:** Performance on different benchmarks regarding general capabilities of language models.

Table 3 presents the performance of our models on a range of standard benchmarks that assess general language understanding and reasoning abilities. By comparing our trained models with the original models, we find that our training method preserves, and in some cases slightly improves, general language and reasoning abilities across diverse benchmarks. Specifically, we observe a consistent improvement in the logical reasoning task (BBH). These results confirm that our approach enables domain-specific improvements without sacrificing overall language model capabilities.

## C   Preliminary Study of DeepSeek-R1 Performance on Game

Recent investigations and preliminary experiments have demonstrated that DeepSeek-R1 (DeepSeek-AI et al., 2025), leveraging its powerful logical reasoning capabilities, can effectively integrate knowledge about the MOBA game, acquired from publicly available textual data. This knowledge includes hero skills, game strategies, equipment information, and more. As a result, DeepSeek-R1 exhibits strong in-game analytical abilities, which have shown promising improvements in existing business applications.

Despite these strengths, DeepSeek-R1 faces two critical challenges: (1) **Efficiency limitations:** While a certain model scale is necessary for a general reasoning model to generalize its capabilities to the HoK domain, the large size of DeepSeek-R1 hinders its practical deployment. The computational cost impacts real-time user experience, making it difficult to guarantee responsiveness in live scenarios; (2) **Performance ceiling:** The model's analytical power fundamentally relies on the combination of data and strategy guides, rather than a deep understanding of the underlying game mechanics. Human-authored guides often omit implicit knowledge; for example, a strategy might advise "avoid pushing the lane too far," but the precise definition of "too far" requires experiential understanding gained through gameplay. Although prompt engineering can inject some additional game mechanic information to partially compensate for this gap, it does not fundamentally enhance the model's reasoning ability within the HoK environment.

Motivated by these limitations, this work aims to develop a lightweight reasoning model tailored for HoK that approaches or even surpasses the reasoning capabilities of DeepSeek-R1. Our objectives are twofold: to reduce computational costs and to achieve a deeper, more intrinsic understanding of game mechanics than general-purpose reasoning models. This approach promises both improved efficiency and enhanced analytical performance in the HoK domain.

## D   Formalization of Reinforcement Learning using GRPO

**Motivation for GRPO.**   Traditional RL algorithms such as PPO (Schulman et al., 2017) have demonstrated effectiveness in language model fine-tuning, but often struggle with high-variance rewards and inefficient credit assignment when applied to long, structured outputs. GRPO addresses these challenges by leveraging group-wise relative advantages, which normalize rewards within a batch of generated completions. This approach not only stabilizes training but also encourages the model to generate responses that are comparatively better within each group, aligning well with the competitive and multi-agent nature of MOBA games.

**Differences from Standard GRPO.**   While our approach is based on the original GRPO framework (Shao et al., 2024), we introduce several adaptations to better suit the MOBA reasoning task. First, we employ a rule-based, binary reward function tailored to the correctness of macro-level

action predictions, rather than relying on human preference or neural reward models. Second, we restrict the action space to a finite set of interpretable strategies, which simplifies reward computation and evaluation. Finally, we omit format-based rewards, as our model demonstrates strong structural adherence by design. These modifications ensure that the RL signal is both reliable and directly aligned with the objectives of strategic reasoning in MOBA games.

| Error Type | Definition |
|---|---|
| Basic Game Knowledge Errors | Errors caused by misunderstanding core game mechanics or roles. E.g., the Support role should prioritize protecting allies and not take Buffs, but the model suggests taking the Buff. |
| Game State Misinterpretation | LLM fails to extract or interpret key battlefield information correctly (e.g., ally/enemy confusion, incorrect HP assessment). E.g., A team fight is occurring in our Blue Buff zone, but the model identifies it as the enemy's Blue Buff zone. |
| Critical Event Oversight | LLM fails to detect or respond to major in-game events (e.g., team fights, High Ground pushes, Dragon/objective contests). E.g., A team fight is happening nearby in Bot Lane, but the model focuses on defending Mid Lane instead. |
| Situational Misjudgment | Poor overall game-state assessment, prioritizing minor/local information over critical objectives. E.g., When the team should push the Base, the model chooses to push an Outer Turret because one enemy hero is low HP. |
| Spatio-Temporal Miscoordination | Inability to judge distances between map zones or positioning accurately, leading to inefficient pathing/decisions. E.g., Bot Lane is far away, yet the model decides to kill the Bot Lane Bird creep instead of the closer Spatial Spirit. |

**Table 4: Definition** of the Common Error Types

```
{                                                 "red": {
  "gameTime": 720,                                  "kills": 12,
   "teams": {                                       "dragons": 1,
   "blue": {                                        "towers": {
    "kills": 15,                                      "destroyed": 2,
    "dragons": 2,                                     "remaining": ["Top Lane First Tower", "Top Lane
    "towers": {                                   Second Tower", "Mid Lane Second Tower",
     "destroyed": 3,                              "Bottom Lane Second Tower", "High Ground
     "remaining": ["Top Lane Second Tower", "Mid  Tower", "Nexus Turret"]
Lane Second Tower", "Bottom Lane First Tower",     },
"Bottom Lane Second Tower", "High Ground           "players": [
Tower", "Nexus Turret"]                             {
    },                                                "name": "Player6",
    "players": [                                      "hero": "Lu Ban Qi Hao",
    {                                                 "level": 11,
      "name": "Player1",                              "kills": 3,
      "hero": "Li Bai",                               "deaths": 4,
      "level": 12,                                    "gold": 7800,
      "kills": 5,                                     "items": ["Endless War Blade"],
      "deaths": 2,                                    "isAlive": false
      "gold": 8500,                                 }, ...
      "items": ["Shadow War Axe", "Tears of Blood    ]
Blade"],                                           }
      "isAlive": true                             }
    },                                            }
    ...
    ]
  },
```

**Figure 5:** Demonstration of JSON object for each game state.

# E  LIMITATIONS AND FUTURE WORK

**Limitations.** Despite the promising results of the TiG framework, there are some limitations that need to be acknowledged:

- **Dependence on LLM Quality:** The effectiveness of TiG is inherently tied to the capabilities of the underlying LLM backbones. Limitations in language understanding or generation, particularly in highly complex or real-time environments, may restrict policy performance.

| Category | id | Action | Explanation |
|---|---|---|---|
| None | 0 | None | No Action triggered for a short period |
| Dragon | 1 | Lord | Deal damage to the Lord (Main Dragon) |
| | 2 | Tyrant | Deal damage to the Tyrant (Early Game Dragon) |
| | 3 | Dragon King | Deal damage to the Dragon King (Late Game Dragon) |
| Tower | 4 | Crystal | Deal damage to enemy Crystal (Nexus) |
| | 5 | Top Tower | Deal damage to Top Lane Tower |
| | 6 | Mid Tower | Deal damage to Mid Lane Tower |
| | 7 | Bot Tower | Deal damage to Bottom Lane Tower |
| Defense | 8 | Defend Crystal | Defend our Crystal |
| | 9 | Defend Top Tower | Defend Top Lane Tower |
| | 10 | Defend Mid Tower | Defend Mid Lane Tower |
| | 11 | Defend Bot Tower | Defend Bottom Lane Tower |
| Hero | 12 | Top Hero | Damage enemy heroes in Top Lane |
| | 13 | Mid Hero | Damage enemy heroes in Mid Lane |
| | 14 | Bot Hero | Damage enemy heroes in Bottom Lane |
| | 15 | River Top Hero | Damage enemies in Upper River (including dragon pit) |
| | 16 | River Bot Hero | Damage enemies in Lower River |
| | 17 | Allied Jungle Hero | Damage enemies in our Jungle |
| | 18 | Enemy Jungle Hero | Damage enemies in opponent's Jungle |
| | 19 | Ally High-ground Hero | Damage enemies on our High-ground |
| | 20 | Enemy High-ground Hero | Damage enemies on enemy High-ground |
| Line | 21 | Top Minions | Clear Top Lane minions |
| | 22 | Mid Minions | Clear Mid Lane minions (including super minions) |
| | 23 | Bot Minions | Clear Bottom Lane minions |
| | 24 | Ally High-ground Minions | Clear minions on our High-ground |
| | 25 | Enemy High-ground Minions | Clear minions on enemy High-ground |
| Buff | 26 | Allied Red | Take our Red Buff |
| | 27 | Enemy Red | Steal enemy Red Buff |
| | 28 | Allied Blue | Take our Blue Buff |
| | 29 | Enemy Blue | Steal enemy Blue Buff |
| Jungle | 30 | Allied Camps | Clear our Jungle camps (non-buff) |
| | 31 | Enemy Camps | Invade enemy Jungle camps |
| | 32 | Void Spirit (Top Crab) | Kill Void Spirit (River objective) |
| | 33 | Crimson Raptor (Bot Crab) | Kill Crimson Raptor (River objective) |
| Grouping | 34 | Top Grouping | Group in Top Lane |
| | 35 | Mid Grouping | Group in Mid Lane |
| | 36 | Bot Grouping | Group in Bottom Lane |
| | 37 | River Top Grouping | Group in Upper River |
| | 38 | River Bot Grouping | Group in Lower River |
| | 39 | Allied Jungle Group | Group in our Jungle |
| | 40 | Enemy Jungle Group | Group in enemy Jungle |
| | 41 | Ally High-ground Group | Group on our High-ground |
| | 42 | Enemy High-ground Group | Group on enemy High-ground |
| Recall | 43 | Recall | Hero at fountain (including walk-back) |

**Table 5:** Action Category Definition.

- **Domain Generalization:** Our current experiments are primarily conducted within digital game environments. The generalizability of TiG to other interactive domains—such as robotics or real-world tasks—remains to be thoroughly investigated.

- **Sample Efficiency:** Although TiG improves sample efficiency compared to baseline methods, it still requires a substantial amount of environment interaction. This requirement may be prohibitive in scenarios where data collection is expensive or time-consuming.

- **Interpretability of Policies:** The interpretability of language-based policies depends on the clarity and faithfulness of the generated explanations. In some cases, these explanations may not fully or accurately reflect the underlying decision-making process.

**Future Works.** Several directions can be explored to improve upon TiG:

- **Scaling and Generalization:** Future work will focus on scaling TiG to a broader range of environments, including those with greater complexity and diversity. Additionally, we aim to enhance the fidelity of generated explanations and incorporate multimodal feedback (e.g., visual or auditory cues) to support richer procedural learning.

- **Long-Term Reasoning:** Another promising direction is to investigate tasks that require long-term memory or reasoning across extended state transitions. Addressing such challenges will require more sophisticated mechanisms for temporal abstraction and memory management.

## F  MORE CASES

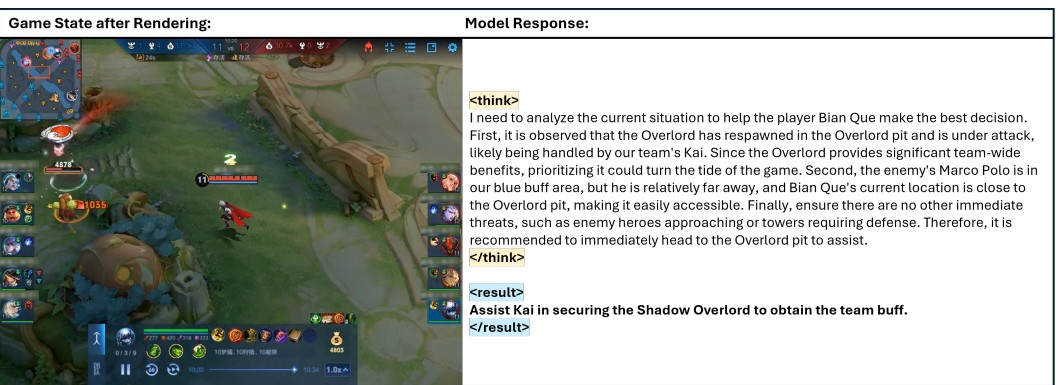

**Figure 6:** One of the cases of TiG. <think> </think> refers to the thinking process of model output, and <result> </result> refers to the model guidance to the main player in natural language.

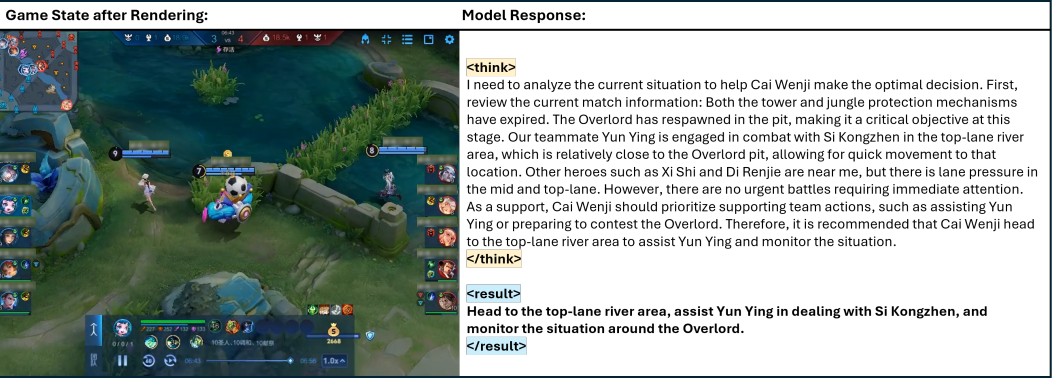

**Figure 7:** One of the cases of TiG. <think> </think> refers to the thinking process of model output, and <result> </result> refers to the model guidance to the main player in natural language.

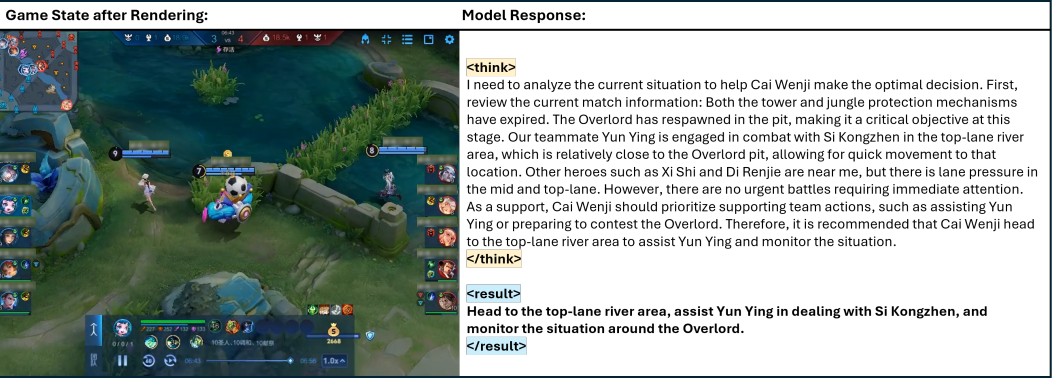

**Figure 8:** One of the cases of TiG. <think> </think> refers to the thinking process of model output, and <result> </result> refers to the model guidance to the main player in natural language.

## G  LARGE LANGUAGE MODELS(LLMS) USAGE DECLARATION

In the preparation of thie work, the authors used DeepSeek-R1 to assist with language polishing and to translate certain content related to the HoK Game from Chinese to English. After using this tool, the authors reviewed and edited the content as needed and take full responsibility for the content of the publication.

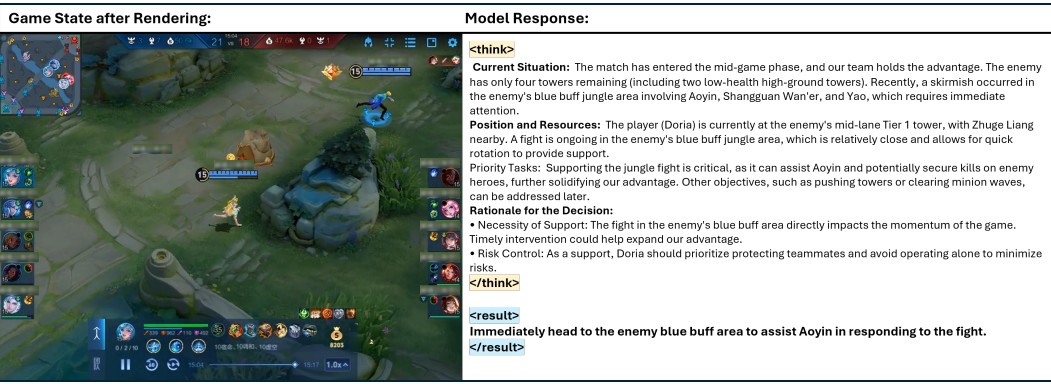

**Figure 9:** One of the cases of TiG. <think> </think> refers to the thinking process of model output, and <result> </result> refers to the model guidance to the main player in natural language.

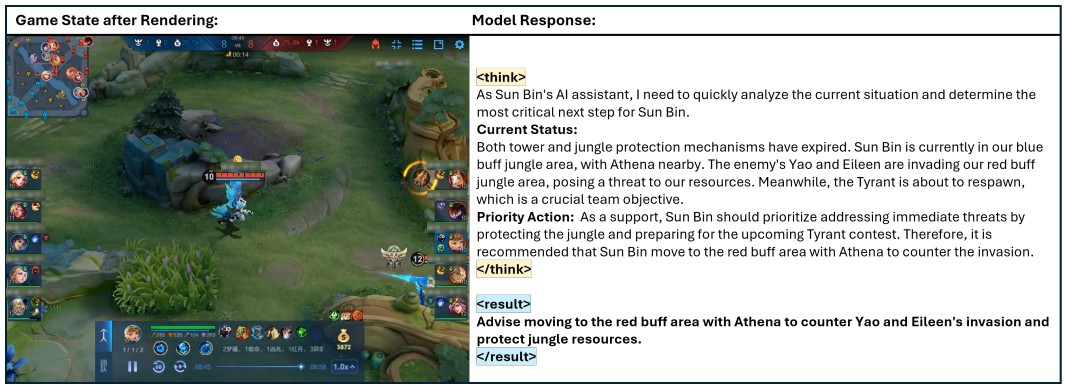

**Figure 10:** One of the cases of TiG. <think> </think> refers to the thinking process of model output, and <result> </result> refers to the model guidance to the main player in natural language.

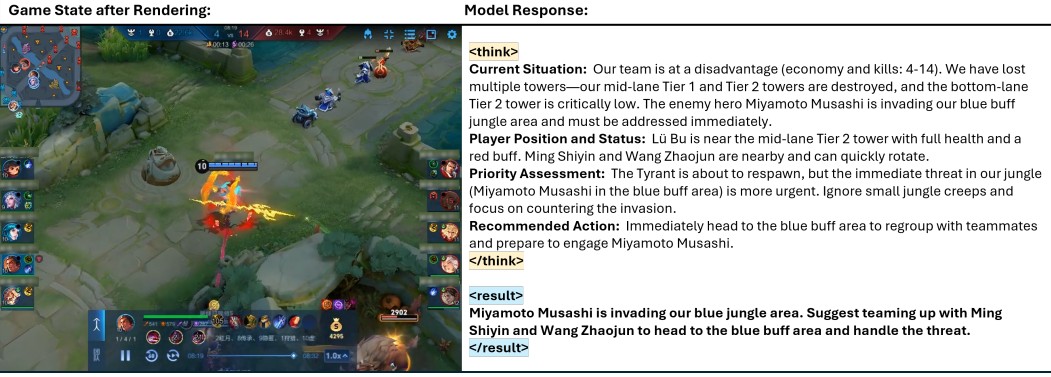

**Figure 11:** One of the cases of TiG. <think> </think> refers to the thinking process of model output, and <result> </result> refers to the model guidance to the main player in natural language.

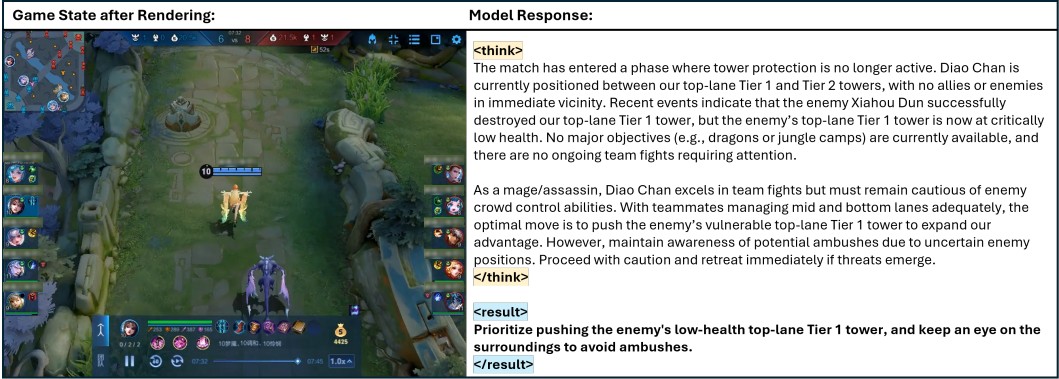

**Figure 12:** One of the cases of TiG. <think> </think> refers to the thinking process of model output, and <result> </result> refers to the model guidance to the main player in natural language.

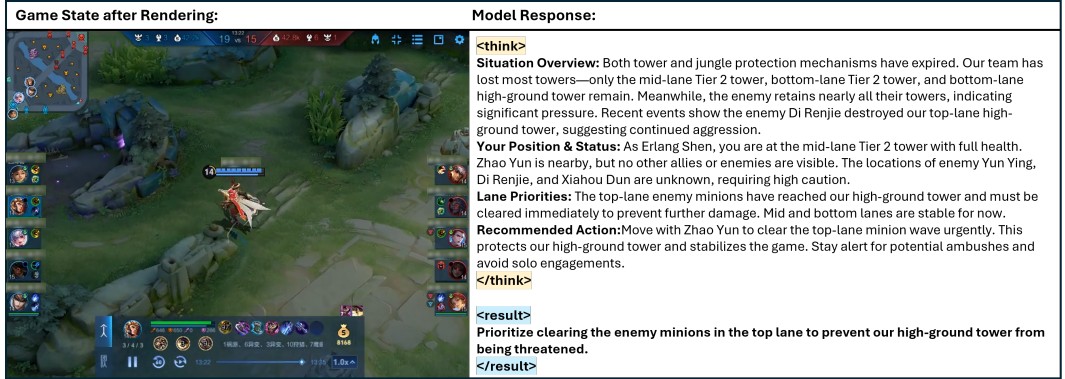

**Figure 13:** One of the cases of TiG. <think> </think> refers to the thinking process of model output, and <result> </result> refers to the model guidance to the main player in natural language.

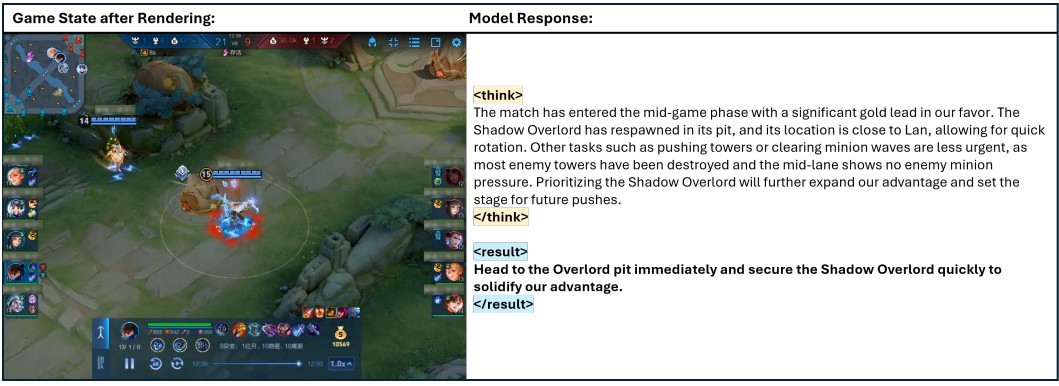

**Figure 14:** One of the cases of TiG. <think> </think> refers to the thinking process of model output, and <result> </result> refers to the model guidance to the main player in natural language.

