# OpenReview forum: "Think in Games: Learning to Reason in Games via Reinforcement Learning with Large Language Models"
_ICLR.cc/2026/Conference — ICLR 2026 Conference Withdrawn Submission_

### Official Review · Reviewer_UMzg · 2025-10-27

**Soundness:** 2
**Presentation:** 2
**Contribution:** 2
**Rating:** 2
**Confidence:** 4

**Summary:**

This paper introduces Think-In-Games (TiG), a framework that bridges the gap between declarative and procedural knowledge in LLMs by training them to make strategic decisions in game environments through reinforcement learning. Specifically, the authors apply a multi-stage training approach combining supervised fine-tuning with Group Relative Policy Optimization to enable LLMs to predict macro-level actions in a game. The paper claims that TiG achieves competitive performance with reduced data requirements while maintaining interpretability through natural language explanations.

**Strengths:**

- The distinction between declarative and procedural knowledge in LLMs is an interesting research direction.
- The paper is well written and easy to follow.

**Weaknesses:**

Please see my detailed questions and concerns below.

**Questions:**

- Why choose the "one frame per minute" sampling strategy? What is the sensitivity of your results to this sampling rate? Could important strategic transitions be missed with such coarse temporal resolution?
- What is the exact distribution of skill levels in your dataset? The paper mentions "above a skill threshold" but provides no quantitative details about player ratings, percentiles, or skill diversity. How does this affect model generalization?
- Did you validate the relabeling algorithm against ground truth expert annotations? What is the gap between your automated relabeling and actual expert judgments?
- How generalizable is this action space design to other games? The entire framework appears tightly coupled to HoK's specific mechanics.
- Can the framework be used in different game genres? If so, additional evaluation experiments across diverse genres should be included. If not, I would suggest the authors narrow the scope of their claim to something like "Think in MOBA Games" instead of "Think in Games."
- How do you handle cases where multiple actions could be equally valid? Games often have multiple viable strategies, but your reward seems only credit exact matches.
- Why no objective in-game performance metrics? Win rate, game duration, objective completion rate, or other measurable outcomes would provide stronger evidence.
- Why no comparison with imitation learning/behavioral cloning?
- How do you validate that the generated reasoning is faithful to the model's actual decision process? Post-hoc rationalization is a well-known problem in AI explainability.

**Details Of Ethics Concerns:**

- How were the anonymized game records collected? What consent mechanisms were used? Did players agree and were they aware that their gameplay would be used for ML training?

---

### Official Review · Reviewer_wXie · 2025-10-31

**Soundness:** 2
**Presentation:** 2
**Contribution:** 1
**Rating:** 2
**Confidence:** 4

**Summary:**

The paper proposes Think-in-Games (TiG), a framework that recasts reinforcement learning (RL) as language modeling for macro-strategy in a MOBA setting, especially Honor of Kings. Given a game state, an LLM outputs a macro-action and a natural-language rationale, then is optimized with GRPO using a binary, rule-based reward that matches human action labels. Reported results show sizable gains in action-prediction accuracy for 14–32B models that rival or surpass larger baselines while maintaining scores on general benchmarks. The paper argues TiG yields procedural competence with interpretable rationales at lower data and compute cost than traditional RL, though evidence relies on action-match metrics within a single MOBA domain.

**Strengths:**

**S1. Practical problem setting and motivating examples**

The paper targets a realistic MOBA macro-strategy setting where latency and partial observability make micro-control impractical. The state is serialized as JSON, and decision outputs are confined to a finite, interpretable macro-action set, which yields a clear interface to downstream controllers or human players. The priority-aware relabeling densifies sparse replay annotations, making frame-level supervision feasible. The case study demonstrates end-to-end usability: the model interestingly produces a concise macro directive with an accompanying rationale that is directly actionable in play.

**S2. Empirical effectiveness of training.**

The multi-stage pipeline (SFT + GRPO) shows consistent accuracy gains over base LLMs across sizes, with some smaller models approaching or surpassing much larger baselines under the paper’s label-matching metric. The training dynamics analyses (e.g., reward and response-length trends) and additional evaluations on general benchmarks indicate that domain fine-tuning does not catastrophically degrade broad capabilities. Human expert assessments on HoK-style tasks further suggest improved decision quality relative to base checkpoints.

**Weaknesses:**

**W1. Unclear Problem Framing**

The paper motivates a declarative–procedural gap and emphasizes “step-by-step explanations,” but the method optimizes macro-action selection with generated rationales and does not specify a formal interpretability objective or a hierarchical control design. The manuscript should state a single primary goal and align claims, method, and evaluation to that goal.

**W2. Weak Justification for Using RL**

The use of RL is not sufficiently justified. The reward is a binary, rule-based match to replay actions, and the task is framed as single-step macro-action prediction. It is unclear what benefits RL provides beyond SFT in principle.

**W3. Limited Generality of Proposed Problem**

The problem and experiments are confined to a single MOBA, **Honor of Kings**, rather than multiple MOBA benchmarks or diverse interactive environments; empirical evidence is presented largely as a focused case study. The case-study focus does not provide the generality needed to justify the work as academically impactful research.

**W4. Limited Novelty of Proposed Method**

The contribution is the simple application of GRPO to macro-action classification with a fixed, interpretable action set and rule-based rewards. These choices are reasonable but do not constitute a substantive algorithmic advance over GRPO or standard supervised approaches.

**W5. Insufficient Details for Reproducibility**

The paper fails to provide sufficient information to ensure reproducibility. The paper omits crucial details such as code, dataset access, a full list of hyperparameters, and a reproducibility statement. Furthermore, the use of inconsistent training steps across models hinders fair comparison and validation of the results.

**Questions:**

**Q1.** The introduction presents interpretability and procedural reasoning as co-primary goals. Which of these is the main scientific contribution?

**Q2.** GRPO is an existing algorithm, and the contribution appears limited to its application within a predefined, hand-crafted macro-action reasoning setting. Could the authors clarify what they consider the key methodological contribution beyond this adaptation?

**Q3.** Could the authors list the specific hyperparameter values used for the GRPO and SFT stages to ensure reproducibility?

---

### Official Review · Reviewer_UoMC · 2025-11-01

**Soundness:** 3
**Presentation:** 3
**Contribution:** 3
**Rating:** 6
**Confidence:** 3

**Summary:**

This paper introduces Think-In-Games (TiG), a framework designed to bridge the gap between the declarative knowledge of LLMs and the procedural knowledge required for effective decision-making in interactive environments. The core idea is to reformulate RL as a language modeling task. An LLM-based policy model takes a structured game state (from the MOBA game "Honor of Kings") as input and generates both a macro-level action and a step-by-step natural language reasoning chain. This policy is then refined using online RL, specifically the GRPO algorithm, with a binary reward signal based on action correctness. The authors demonstrate that TiG enables smaller models (e.g., 14B parameters) to achieve competitive or superior performance compared to much larger models (e.g., DeepSeek-R1 with 671B parameters) on in-game action prediction and question-answering tasks, while also providing interpretable reasoning traces.

**Strengths:**

1. The paper clearly identifies a critical and under-explored gap in AI: the disconnect between LLMs' static knowledge and the dynamic, procedural knowledge required for interaction.
2. TiG has a clear interpretability by requiring the model to generate a reasoning chain for every decision.
3. The results are impressive. Demonstrating that a 14B parameter model can surpass a 671B parameter model in a specific domain is a powerful claim that highlights the data and compute efficiency of the proposed method. The multi-stage training (SFT + GRPO) is shown to be highly effective.
4. The paper includes a comprehensive evaluation, including main results, ablation studies, error analysis, and qualitative case studies.

**Weaknesses:**

1.   The binary reward is simplistic. It assumes there is only one correct action per state and heavily relies on the quality of the relabeled expert data. This reward does not capture the quality of the reasoning or the nuanced long-term value of strategic decisions that may not have an immediate, observable counterpart in the expert's action. It risks the model simply learning to imitate the dataset rather than truly understanding the game.
1.   The claim of generalizability is primarily supported by performance on a related QA task (TiG-QA) within the same game. There is no evidence provided that the acquired "procedural understanding" or the TiG framework itself generalizes to other, distinctly different games or interactive domains.

**Questions:**

1.   Given the non-trivial inference time of LLMs, what is the estimated time to generate a single <think>/<answer> output?
2.   Did you experiment with more sophisticated reward functions? Why was the simple binary reward ultimately chosen, and how do you mitigate the risk of the model overfitting to the potentially imperfect expert labels?
3.   How can you verify that the generated reasoning is a genuine reflection of the model's decision process and not just a post-hoc justification? Have you conducted ablations where you suppress the reasoning generation during training or inference to see its direct impact on action selection accuracy and policy learning?
4.   The paper claim TiG is a general framework. What specific experiments can be proposed to test its transferability to a fundamentally different game genre? Would the action space and state representation need to be completely re-engineered?

---

### Note · Authors · 2025-11-24

**Comment:**

The authors have decided to withdraw this manuscript from the review process to conduct further internal revisions and reorganization of the work.  We would like to thank the reviewers and area chairs for their time and feedback.

**Withdrawal Confirmation:**

I have read and agree with the venue's withdrawal policy on behalf of myself and my co-authors.